# The Biological Effect of Platelet-Rich Plasma on Rotator Cuff Tears: A Prospective Randomized In Vivo Study

**DOI:** 10.3390/ijms25147957

**Published:** 2024-07-21

**Authors:** Charalampos Pitsilos, Sofia Karachrysafi, Aikaterini Fragou, Ioannis Gigis, Pericles Papadopoulos, Byron Chalidis

**Affiliations:** 12nd Orthopaedic Department, Aristotle University of Thessaloniki, 54635 Thessaloniki, Greece; chpitsilos@outlook.com (C.P.); jgigis71@gmail.com (I.G.); perpap@otenet.gr (P.P.); 2Research Team “Histologistas”, Interinstitutional Postgraduate Program “Health and Environmental Factors”, Department of Medicine, Faculty of Health Sciences, Aristotle University of Thessaloniki, 54124 Thessaloniki, Greece; sofia_karachrysafi@outlook.com; 3Laboratory of Histology-Embryology, Department of Medicine, Faculty of Health Sciences, Aristotle University of Thessaloniki, 54124 Thessaloniki, Greece; 4Laboratory of Biological Chemistry, Medical Department, School of Health Science, Aristotle University of Thessaloniki, 54124 Thessaloniki, Greece; katerinafragou@hotmail.com; 51st Orthopaedic Department, Aristotle University of Thessaloniki, 57010 Thessaloniki, Greece

**Keywords:** platelet-rich plasma, tendon, rotator cuff, fibroblast, collagen, histology, microscopy

## Abstract

The positive effect of platelet-rich plasma (PRP) on tendon metabolism has been extensively investigated and proven in vitro. Additionally, in vivo animal studies have correlated the application of PRP with the enhancement of tenocyte anabolic activity in the setting of tendon degeneration. However, less is known about its in vivo effect on human tendon biology. The purpose of the current prospective randomized comparative study was to evaluate the effect of PRP on torn human supraspinatus tendon. Twenty consecutive eligible patients with painful and magnetic resonance imaging (MRI)-confirmed degenerative supraspinatus tendon tears were randomized in a one-to-one ratio into two groups. The patients in the experimental group (*n* = 10) underwent an ultrasound-guided autologous PRP injection in the subacromial space 6 weeks before the scheduled operation. In the control group (*n* = 10), no injection was made prior to surgery. Supraspinatus tendon specimens were harvested from the lateral end of the torn tendon during shoulder arthroscopy and were evaluated under optical and electron microscopy. In the control group, a mixed cell population of oval and rounded tenocytes within disorganized collagen and sites of accumulated inflammatory cells was detected. In contrast, the experimental group yielded abundant oval-shaped cells with multiple cytoplasmic processes within mainly parallel collagen fibers and less marked inflammation, simulating the intact tendon structure. These findings indicate that PRP can induce microscopic changes in the ruptured tendon by stimulating the healing process and can facilitate a more effective recovery.

## 1. Introduction

The rotator cuff (RC) surrounds the glenohumeral joint of the shoulder and consists of four muscles: subscapularis, supraspinatus, infraspinatus and teres minor [1]. The subscapularis is inserted to the lesser tuberosity, while the other three to the greater tuberosity [2]. The tendon-to-bone insertion site is divided into four zones: 1. Tendon midsubstance, 2. Fibrocartilage, 3. Mineralized fibrocartilage and 4. Bone [3]. There is a gradual and continuous change in composition from zones 1 to 4; zone 1 is composed primarily of well-aligned collagen types I and XII and spindle-shaped cells, while zone 4 contains randomly oriented fibers mostly of collagens II, IX, and X and round-shaped cells [4]. 

Tendons are connective tissues composed of cells, extracellular matrix, vessels and nerves [5]. Tenocytes, also called tendon fibroblasts, are the main cells within the tendon, but progenitor stem cells are also present [6]. Tenocytes display an elongated appearance with a complex network of cytoplasmic processes that links adjacent cells [7]. The extracellular matrix includes water, collagen, proteoglycans, glycoproteins and other molecules [8]. The most abundant collagen type in healthy tendons is type I, followed by type III [9]. 

Contemporary microscopy techniques can facilitate a thorough evaluation of the morphology of tenocytes and the structure of tendon tissue [10]. The optical or light microscope has been effectively used for the observation of cell morphology and extracellular matrix structure [11]. For this purpose, different staining techniques have been described, including hematoxylin and eosin, Van Gieson, toluidine blue, Picrosirius and Xylidine Ponceau stains [12,13,14]. The Movin and the Bonar scoring systems, which assess the tenocyte and extracellular matrix properties, have mainly been used for the classification of the histopathological findings of tendons in optical microscopy [15]. Electron microscope analysis is suitable for the evaluation of smaller components, such as the nuclei, cytoplasm, collagen fibers or fibrils and vessels [16,17].

The pathophysiology of an RC tear is a complex issue beginning with the development of degenerative changes characterized by matrix disorganization and an inflammatory response that is closely associated with mechanical overloading [18]. The pathological changes within the tendon substance include a decrease in cells number with a higher apoptosis rate, more round-shaped appearance of tenocytes and an increase in the content of proteoglycans and disorganized collagen fibers [19]. These structural changes alter the biological and biomechanical properties of the tendon, causing sequential progression to a partial-thickness or full-thickness tear [20]. Traumatic rupture of an otherwise healthy tendon has also been described; however, whether this is traumatic or degenerative in nature is controversial and a matter of debate [21].

Platelet-rich plasma (PRP) is an autologous blood-derived product that contains an excessive concentration of platelets along with platelet-released growth factors and cytokines [22]. The number of platelets in PRP is greater or equal to 5.5 × 10^10^ per 50 mL and is approximately two to seven times higher than that in whole blood [23]. Based on the preparation method, it may or may not contain leukocytes and then it is classified as leukocyte rich-PRP (LR-PRP) or leukocyte poor-PRP (LP-PRP), respectively [24]. While the optimal platelet and leukocyte concentration is still debated, PRP’s therapeutic effect has become increasingly popular for the treatment of various musculoskeletal disorders, including lateral epicondylitis, rotator cuff and Achilles tendinopathy, hip and knee osteoarthritis and plantar fasciitis [25,26]. Additionally, and due to its potential anti-inflammatory and regenerative effect, it has been successfully applied as adjuvant treatment to different orthopedic procedures, such as shoulder arthroscopy and total knee arthroplasty [27,28]. 

Platelet-rich plasma holds immense promise for enhancing tendon tissue healing in patients with tendon injuries and ruptures. When injected at the site of tendon pathology, it can recapitulate the natural biologic process of tendon healing and repair [29]. The biologically active agents contained therein exert both anti-inflammatory and pro-inflammatory effects on the tenocytes [30]. In tendinopathy, which is an inflammatory condition, the application of PRP may promote tendon healing as it stimulates tenocyte migration and proliferation, improves tissue vascularization and increases collagen deposition [31]. In the absence of inflammation, PRP seems to act in two ways: firstly, by the activation of pro-inflammatory tumor necrosis factor α and nuclear factor kappa-light-chain-enhancer of activated B cells pathways, and secondly, by stimulating genes related to cellular proliferation and tendon collagen remodeling [32]. 

The outcome of PRP application on RC tendon tears has been thoroughly evaluated in laboratory and clinical settings [33,34,35]. In vitro studies demonstrated the positive effect of PRP on cell proliferation, extracellular matrix gene expression and collagen synthesis of tenocytes derived from donors with degenerative RC tendon tears [36,37,38]. The injection of PRP at the site of an RC tear was related to improved long-term outcome in terms of pain and shoulder function [39]. Additionally, PRP supplementation during RC repair has also been associated with both improved clinical outcomes and decreased re-tear rates [40,41].

The purpose of the current prospective randomized comparative study was to evaluate the in vivo effect of PRP injection on tendon structure and tenocyte morphology in patients with degenerative supraspinatus tendon tear. We hypothesized that preoperative PRP injection in the subacromial space would be correlated with more well-organized tendon tissue and the healthier appearance of tenocytes at the time of surgical intervention compared to controls. 

## 2. Results

### 2.1. Demographics

Ten patients were enrolled in each group. Patient demographics are summarized in Table 1. The two groups did not differ in respect to age, gender distribution, duration of symptoms and visual analogue scale (VAS) score for shoulder pain. The latter was evaluated before PRP injection in the experimental group and preoperatively in the control group. Interestingly, in the experimental group, the VAS score was significantly reduced from 6.6 at the time of PRP injection to 1.4 at the day before surgery (*p* < 0.05).

### 2.2. Optical Microscopy Evaluation

In the experimental group, oval tenocytes were found to be abundant and well organized within regularly arranged collagen fibers. No signs indicative of extended inflammation were detected. The Bonar score ranged from 0 to 1 with a mean score of 0.4. In the control group, the tenocytes yielded nearly normal appearance and were less organized within more irregularly arranged collagen fibers compared to the experimental group. The presence of areas of accumulated inflammatory cells indicated a more extended inflammation reaction. The mean value of the Bonar score was 3 (range; 2–4) (Table 2) (Figure 1, Figure 2 and Figure 3).

### 2.3. Electron Microscopy Evaluation 

In the experimental group, the collagen fibers and fibrils were found to be well organized. Tenocytes were numerous with multiple cytoplasmic processes (Figure 4). In the control group, the collagen fibers and fibrils were disorganized with disrupted architecture of the interstitial tissue. Tenocytes were identified at lower frequency and their cytoplasmic processes appeared shorter compared to the experimental group. (Figure 4 and Figure 5).

## 3. Materials and Methods

All procedures described in this study were approved by the Ethics Committee of our hospital and university (No. 15043—27 September 2023). All patients gave written informed consent to participate in the study.

### 3.1. Study Design

This study presents a single-blinded randomized clinical trial. Twenty consecutive eligible patients were randomized in a one-to-one ratio into two groups: the experimental group (*n* = 10) and the control group (*n* = 10). According to the power analysis based on previous literature data, a total of at least sixteen samples were sufficient to reach statistically valid conclusions. The patients of the experimental group underwent an ultrasound-guided autologous PRP injection in the subacromial space 6 weeks before the scheduled operation. In the control group, no injection was made prior to surgery. Supraspinatus tendon specimens were harvested from the lateral end of the torn tendon during shoulder arthroscopy. These specimens were properly prepared for examination under the optical and electron microscope. The pathologist evaluated each sample blindly and the results were collected, analyzed and presented accordingly. 

### 3.2. Patient Selection

Patients with a painful and magnetic resonance imaging (MRI)-confirmed degenerative supraspinatus tendon tear, who would be treated with arthroscopic repair, were deemed eligible to participate in the current study. The inclusion criteria were as follows: (1) patients with full-thickness supraspinatus tear type 1 or 2 according to DeOrio and Cofield classification system [42]; (2) persistent symptoms after conservative management of at least 3 months; and (3) aged between 40 and 70 years. Patients with shoulder MRI findings of grade 3 or 4 fatty infiltration of the supraspinatus muscle according to Fuchs classification [43]; history of injection around the shoulder joint during the past 12 months; blood platelet count of less than 150.000/mm^3^; or history of former shoulder surgery were excluded from the study.

### 3.3. Preparation of PRP

The PRP was prepared using the TriCell PRP M Blood Separation Kit (REV-MED, Seongnam, Republic of Korea) according to the manufacturer’s instructions. In the experimental group, twenty-seven milliliter of blood was taken from each patient, using a syringe prefilled with 3ml of heparin sodium (5000 IU/mL). The mixture was placed into the collection tube and centrifuged for 5 min at 3200 revolutions per minute, until red blood cells were separated from plasma and buffy coat. Subsequently, a second centrifugation of the mixed plasma and buffy coat was performed and, finally, 3–4 mL of leukocyte-poor PRP was obtained. Based on manufacturer’s information, approximately 3 to 5 billion platelets were delivered to the subacromial space during each injection. 

### 3.4. Injection of PRP 

In the experimental group, a PRP injection was administered six weeks prior to scheduled arthroscopic surgery. This time point was selected according to the published results of observational studies, where clinical improvement was evident even at six weeks after shoulder PRP injection in cases of chronic rotator cuff tears [44]. During the preoperative intervention, the PRP fluid was injected into the subacromial space of the examined shoulder under ultrasound guidance, in a lateral-to-medial direction (Figure 6). 

### 3.5. Tendon Sample Harvesting

During arthroscopy, a full-thickness supraspinatus tendon specimen sized approximately 3 × 5 mm was harvested from the lateral edge of the tendon tear using a basket punch (Figure 7). The sample was divided into two equal parts. One part, which would be examined under optical microscope, was embedded in sterile 10% formalin solution (out of 35% formaldehyde stock solution). The second part, which would be evaluated by electron microscopy, was sectioned into <0.5 cm^3^ pieces and placed in glutaraldehyde solution 3%.

### 3.6. Tendon Sample Preparation for Optical Microcopy

The tendon samples for the optical microscopy examination were removed from the buffered formalin and were dehydrated through an ascending series of alcohol solutions (76%, 96%, 100%, 100%). Subsequently, they were cleared with xylene for four hours and dipped into liquid paraffin for an additional four hours. Then, they were placed in metallic molds, soaked in liquid paraffin and allowed to cool at 4 °C for twenty minutes.

Using a semi-automated microtome, the paraffin blocks were sectioned at a thickness of 3 μm. Eight sagittal sections for each patient were randomly collected and placed on standard microscope slides. All slides were dried initially at room temperature for one hour and then were placed in the oven at 65 °C for one more hour. Afterwards, they were dipped in xylene solution for ten minutes for deparaffinization. Subsequently, a descending series of alcohol solutions (100%, 100%, 96%, 76%) was used for hydration. Sections were then stained with hematoxylin for five minutes and rinsed in tap water for five more minutes. A 1% differentiation solution was used for one second for partial discoloration of hematoxylin. Sections were stained with eosin for one minute, dehydrated in ethanol for five minutes, and cleaned in xylene for another five minutes. Finally, the slides were covered with “Canada balsam” for light microscopical analysis.

### 3.7. Tendon Sample Preparation for Transmission Electron Microscopy

The samples for the electron microscopy examination remained in glutaraldehyde 3% for 2 h. Subsequently, they were fixed in osmium tetroxide (OsO4) 1% for 1 h. They were stained using uranyl acetate 1% for 16 h and then dehydration was performed using ascending ethanol concentrations. Samples were embedded in Epon resin and then ultra-thin sections (50–100 nm) were taken. Finally, sections were stained with Reynolds’ stain. 

### 3.8. Assessment of Tendon Sample

In the optical microscopy scanning, the histopathological assessment included the evaluation of collagen fiber arrangement, cell density and the presence of inflammatory reaction. The Bonar grading system was used as the assessment tool for quantification of tendon morphology. This score considers four parameters: tenocyte morphology, ground substance, collagen architecture and vascularity [45] (Table 3). Each variable is quantified on a scale from 0 to 3 (0 = normal, 1 = slightly abnormal, 2 = abnormal and 3 = markedly abnormal). 

The collagen and tenocyte characteristics were assessed by scanning electron microscopy, using a TEM JEOL 1011 in 80 kV (JEOL-Tokyo, Tokyo, Japan). The collagen examination included the assessment of the orientation and integrity of collagen fibers and fibrils and the appearance of the interstitial tissue. Regarding tenocyte analysis, the cell density and the cytoplasmic characteristics were evaluated. 

### 3.9. Visual Analogue Scale

The visual analogue scale (VAS) score from 0 to 10 was used to assess the severity of shoulder pain at the time of recruitment to the study in both groups. In the experimental group, the VAS score was additionally evaluated on the day of the operation (six weeks after the PRP injection). 

### 3.10. Statistical Analysis 

All the extracted data were transcribed into SPSS (IBM Corp. Released 2017. IBMSPSS Statistics for Windows, Version 25.0. Armonk, NY, USA: IBM Corp.) and subsequently analyzed. Non-parametric variables of the two groups were compared using Mann–Whitney U test. All tests were two-sided and statistical significance was assumed at a *p* value of <0.05. 

## 4. Discussion

The most important finding of this in vivo experimental study was that PRP could positively affect the microscopic characteristics of human supraspinatus tendon in the setting of a degenerative RC tear. The injection of PRP in the subacromial space was correlated with a healthier appearance of tendon tissue after six weeks, including a better orientation of collagen fibers, an increased number of tenocytes and a decrease in inflammatory cells. This organized tendon structure, which resembles an intact tendon, is considered more suitable for successful repair and promises a better functional outcome. 

The histopathological features of the torn RC tendon consist of a loss in structural organization as well as decreased vascularity and fibrosis, which predispose to low healing capacity [46]. Additionally, and due to focal synovitis, inflammatory cells are able to migrate into the tendon tissue through the hyperplastic vessels of the inflamed subacromial bursa [47]. These findings have been confirmed by many studies [48,49]. Hashimoto et al. [50], analyzing the microscopic appearance of the ruptured RC tendon, found significant pathological changes including thin and disoriented collagen fibers, myxoid metaplasia and hyaline degeneration. These degenerative changes are related to chronic repetitive tendon injury and strain leading to microruptures of collagen fibers and differentiation of tendon architecture [51,52]. In contrast, the intact tendon tissue has a different appearance as it contains healthy, abundant oval-shaped tenocytes with multiple cellular processes within parallel collagen fibers [53,54]. In the current study, tendon samples of the torn supraspinatus tendon yielded a loss in structural consistency, with the presence of both oval and rounded tenocytes and randomly dispersed and disoriented collagen fibers. However, the injection of PRP stimulated the tenocyte anabolic activity and improved the tendon biologic tissue response, which was proven by the detection of more oval-shaped cells with elongated cytoplasmic processes and better collagen orientation. These findings, along with the evidence of regression of bursal synovitis and inflammation at the free end of the torn tendon may optimize the postsurgical outcome of tendon repair. 

Platelet-rich plasma has also been associated with improved tenocyte metabolism and function in in vivo and in vitro studies [37,55]. According to the literature, the laboratory experimental procedures include the culturing of tenocytes derived from animal or human tendons in matrix containing PRP [56]. Xu et al. [57] found enhanced tenocyte viability, proliferation and migration compared to controls three days after culturing rat Achilles tendon cells on a PRP–collagen matrix. Additionally, at two weeks, they noticed higher metabolic activity due to the upregulation of the expression of collagen type I (COL1), collagen type III (COL3), scleraxis and tenascin C genes. Similarly, Anitua et al. [58] cultured human semitendinosus tenocytes for 6 days and noticed that the presence of PRP was associated with accelerated cell proliferation and increased synthesis of human procollagen type I C-peptide, compared to controls. Furthermore, they observed a greater secretion of neovascularization agents, including the hepatocyte growth factor (HGF), the transforming growth factor (TGF)–1 and the vascular endothelial growth factor (VEGF). In another study, the same authors investigated the effect of platelet-derived growth factor (PDGF) and TGF-1 on tenocytes derived from human semitendinosus tendon. Cell proliferation was enhanced by the addition of PDGF, while the presence of TGF-1 was related with increased collagen, HGF and VEGF synthesis [59]. 

Using supraspinatus tendon samples from patients with RC arthropathy, Cross et al. [60] found that tenocytes cultured in media containing PRP expressed an increased COL1A1:COL3A1 ratio. In the study of De Mos et al. [61], tenocytes derived from hamstring tendons of children with history of knee contracture were cultured in media containing PRP at concentrations of 0%, 10% or 20%. The cell count and morphology as well as the expression of genes associated with collagen metabolism and vascularization were evaluated after four, seven and fourteen days. Compared to the initial spindle-shaped, fibroblast-like appearance, the tenocytes appeared more stretched with oblong shape on the fourteenth day. Moreover, a positive dose-related effect of PRP on cell proliferation was identified. Regarding growth factors, an increase in the expression of genes associated with in vivo accelerated catabolism of injured tendons and angiogenesis was also detected. In another study, Jo et al. [62] noticed that the addition of calcium-activated PRP in cultured tenocytes derived from degenerated RC, was correlated with the stimulation of cell proliferation and enhancement of total collagen and glycosaminoglycan synthesis but with no changes in cell morphology after a time period of fourteen days.

In in vivo experimental studies, tendon samples were harvested for analysis after PRP injection at the examined area [37]. Zhang et al. [63] investigated the influence of saline, PRP or PRP with inhibition of high-mobility group box 1 injections on the healing process of the ruptured patellar tendon in mice models. At seven days, standard PRP was associated with faster tendon healing and almost typical tendon structure. In another experiment in rats with patellar tendon rupture, Zhang et al. [64] compared the healing effect of injected PRP activated by thrombin or proteinase-activated receptor (PAR) 1 or PAR4. A control group without any PRP intervention was also examined. In the thrombin-PRP group, overgrowth scar-like tissue was apparent at the site of tendon healing after eight weeks. In the PAR4-PRP group, tendon-like tissue with well-organized collagen fibers and very few blood vessels were identified, whereas in the remaining two groups, there were no signs of tendon healing. 

Kobayashi et al. [65] applied PRP hydrogel on rat patellar tendons with full-thickness tears and studied its effect for a period of ten weeks. At two and four weeks, the PRP group showed a higher Bonar score with faster collagen rearrangement, increased blood capillaries, thickening of the tendon and earlier invasion of inflammatory cells compared to controls. This effect was minimized at the later phase of the tendon-repair process until the tenth week. In the study of Matsunaga et al. [66], a PRP–fibrin matrix was used to bridge the gap of rabbit ruptured patellar tendons. A network of dense and longitudinally aligned collagen bundles was identified at the intervention site after 4 weeks. Yu et al. [67] evaluated the healing process of partial transected rat Achilles tendon at five and ten days after PRP injection. Compared to the control, non-injected tendons, PRP was associated with more collagen matrix, less random fibroblast orientation and less ED1+ macrophages at five days. Abundant collagen fibers within a newly formed tendon were found in the PRP group at ten days. The application of PRP was also correlated with decreased cell apoptosis at both time points. 

It is still debated whether LR-PRP or LP-PRP is more effective for the treatment of tendon pathology [68]. In an in vitro study, Lin et al. [69] studied the impact of different compositions of PRP on tenocytes derived from patients with chronic RC tears. The LR-PRP proved to be more effective in enhancing cell proliferation and growth factor release compared to LP-PRP. In contrast, in an in vivo animal study, Nishio et al. [70] applied PRP gel on mice patellar tendons with induced full-thickness tears. The authors found that LP-PRP improved collagen arrangement and promoted tendon healing faster than LR-PRP. Additionally, in a recent systematic review including nine randomized control trials, Peng et al. [71] concluded that LP-PRP as adjuvant to RC repair was associated with improved clinical outcome and reduced re-tear rate compared to LR-PRP. Based on these findings and the potential superiority of LP-PRP over LR-PRP, we elected to choose the option of LP-PRP in our experimental study.

The Bonar score is a commonly used scoring system for the quantification of histological alterations in degenerative tendon [72]. Lundgreen et al. [73] found an increased Bonar score in partially torn supraspinatus tendon samples compared to intact reference tendons. Many modifications of the original version have been described so far but no clear association between score value and the clinical status of patients with RC tears has been recognized [74]. The classic Bonar score was used by Sethi et al. [75] to correlate the microscopic appearance of the torn supraspinatus tendon with the postoperative American Shoulder and Elbow Surgeons functional score in 105 patients with full-thickness tears. The authors discovered that the degree of degeneration was not predictive of the functional outcome of the arthroscopic repair. However, the increased Bonar score was associated with higher VAS values. 

An interesting question is how the injection of PRP into the subacromial space can affect the tendon metabolism, as the subacromial bursa separates these two compartment layers [76]. The proven interaction between the tendon and bursal cells seems to be the main mechanism of action [77]. The subacromial bursa tissue contains abundant intrinsic trophic and pluripotent factors that may influence the biologic properties of RC [78]. Bursal progenitor and fibroblast-type cells can stimulate cytokine signaling, extracellular matrix formation, growth factor release and vascular-response pathways which direct tendon cellular activity in the setting of RC injury [79,80]. However, Muench et al. [81] failed to prove any effect of PRP on adhesion and proliferation of subacromial bursa-derived progenitor cells in vitro. Still, Marshall et al. [82] suggested that bursa could be used for drug delivery to facilitate tendon healing. As there is a paucity of data about the effect of PRP on subacromial bursa biology, the explanation of the findings of the current study lies either in the direct delivery of PRP growth factors from the subacromial space to the underlying tendon or in the secretion of anabolic molecules from the bursal tissue that upregulate the supraspinatus tendon response to achieve healing. 

This study should be considered subject to the following limitations. Firstly, the number of enrolled patients in each group was small. Secondly, the collection of biopsy specimens during PRP injection might have been more accurate for the investigation of the effect of PRP on tendon biology. However, this would be an extra-operative procedure under regional or general anesthesia, which might be also considered unethical. Thirdly, the exact concentration of platelets in each PRP sample was not counted as an additional procedure. We believe that any reduction in the small volume of the manufactured PRP material (3–4 mL) would affect its impact on tendon tissue properties. Finally, there was a lack of ultrasound assessment of the rotator cuff after the PRP injection. Therefore, the evaluation of the effect of PRP on tendon tissue characteristics via high-resolution ultrasound imaging was not performed.

## 5. Conclusions

This in vivo human study evaluated the effect of a subacromial injection of PRP on the microscopic structure of the torn supraspinatus tendon. The findings indicate that the regenerative potential of PRP’s bioactive factors can stimulate tenocyte anabolic activity and enhance the tendon healing process. The tendon appearance six weeks after the application of PRP was quite similar to that of intact tendon, in contrast to the degenerative features of the tendon tissue in the absence of the PRP effect. These findings indicate that PRP may improve the healing capacity of surgically repaired tears of the RC. However, further research is warranted to investigate the in vivo molecular effect of PRP on both tendon tissue and the biological process of tendon–bone integration.

## Figures and Tables

**Figure 1 ijms-25-07957-f001:**
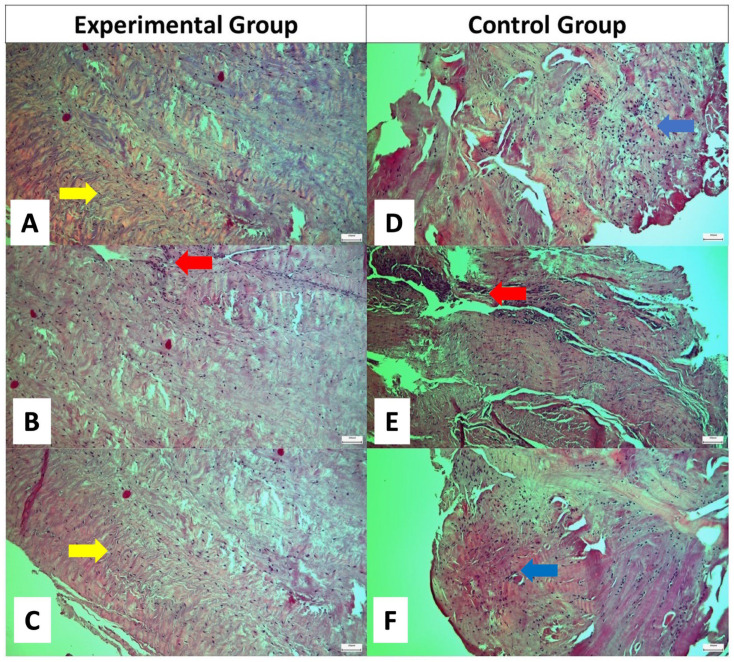
Optical microscopy image of tendon specimens. (**A**–**C**) Specimens from three different patients in the experimental group. (**D**–**F**) Specimens from three different patients in the control group. In the experimental group, abundant tenocytes lie within parallel, waveform collagen fibers (yellow arrows). A few inflammatory cells are also identified (red arrows). In the control group, collagen fibers yield both parallel and disorganized (blue arrows) distribution with numerus interposed tenocytes. Accumulation of inflammatory cells is marked with the blue arrow. Hematoxylin and eosin staining; original magnification ×100.

**Figure 2 ijms-25-07957-f002:**
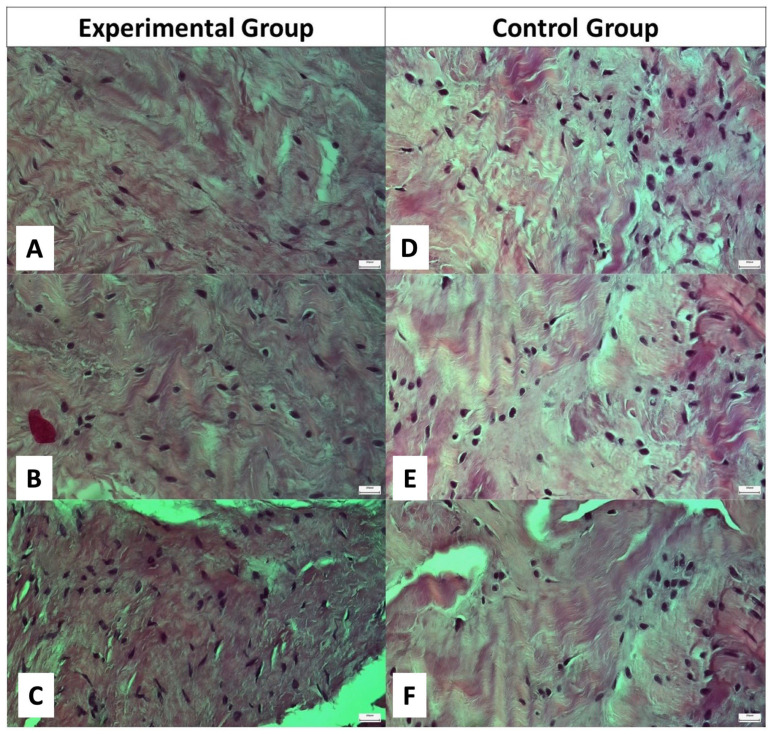
Optical microscopy image of tendon specimens. (**A**–**C**) Specimens from three different patients in the experimental group. (**D**–**F**) Specimens from three different patients in the control group. In the experimental group, oval-shaped tenocytes are identified within parallel, waveform collagen fibers. In the control group, oval and rounded tenocytes lie within disorganized, disoriented collagen fibers. Hematoxylin and eosin staining; original magnification ×400.

**Figure 3 ijms-25-07957-f003:**
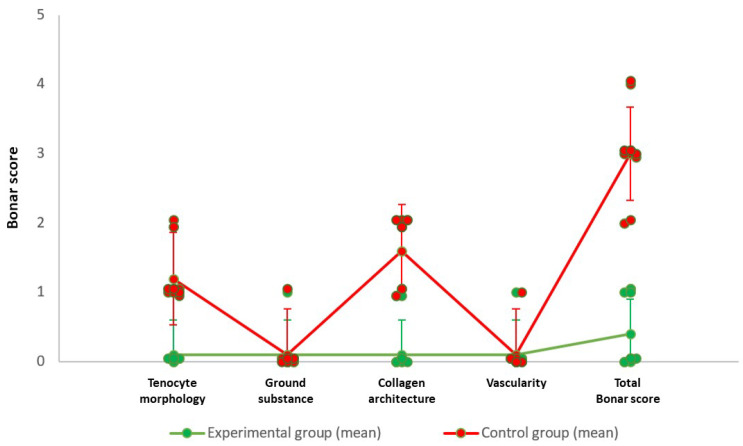
A plot demonstrating the deviation of Bonar score parameters’ grades in the experimental and control groups.

**Figure 4 ijms-25-07957-f004:**
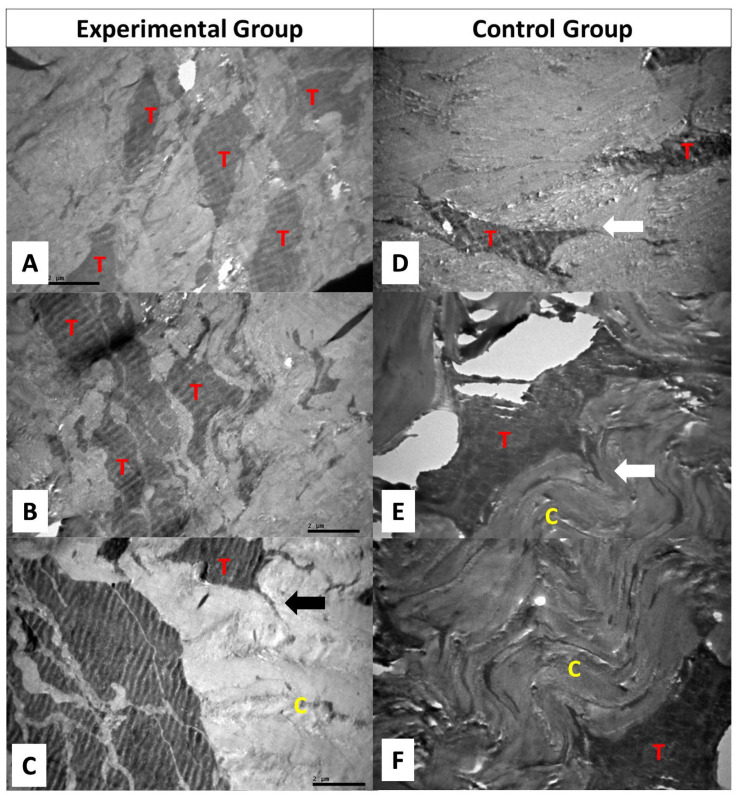
Electron microscopy image of tendon specimens. (**A**–**C**) Specimens from three different patients in the experimental group. (**D**–**F**) Specimens from three different patients in the control group. In the experimental group, many tenocytes (T) with multiple cytoplasmic processes (black arrow) are identified within parallel collagen fibers (C). Original magnification ×6000 (**A**,**B**), ×8000 (**C**). In the control group, collagen fibers appear in a wavy, parallel orientation (C). Tenocytes (T) with short cytoplasmic processes are distinguished (white arrows). Original magnification ×8000 (**A**), ×12,000 (**B**,**C**).

**Figure 5 ijms-25-07957-f005:**
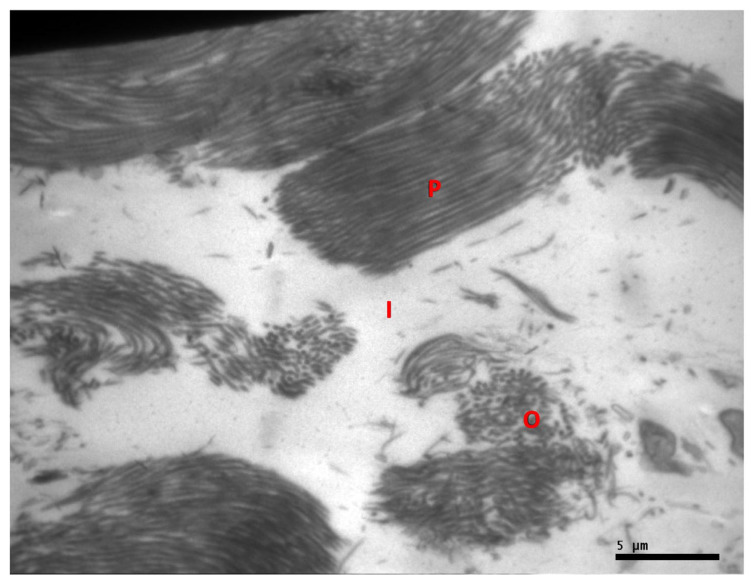
Electron microscopy image of tendon specimen received from the control group. Collagen fibrils with fragmentation are distributed in parallel (P) and oblique (O) orientation. Disruption of the normal architecture of interstitial tissue (I) is identified. Original magnification ×20,000.

**Figure 6 ijms-25-07957-f006:**
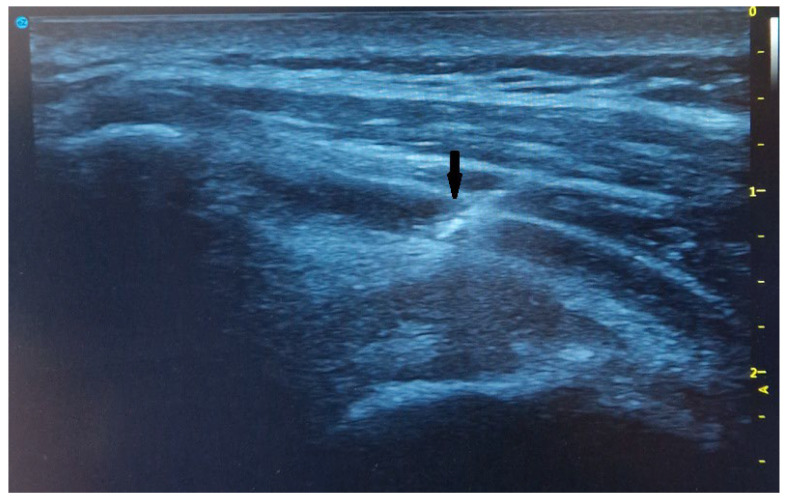
Ultrasound image of the shoulder. After confirmation of the needle tip position in subacromial space (black arrow), PRP injection is carried out.

**Figure 7 ijms-25-07957-f007:**
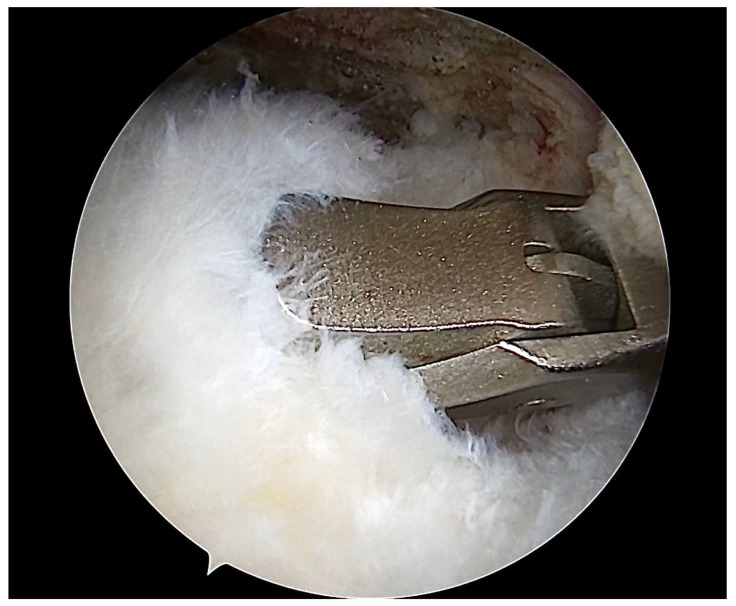
Arthroscopic image of the subacromial space. A basket punch is used to harvest a tendon specimen from the lateral end of the torn supraspinatus tendon.

**Table 1 ijms-25-07957-t001:** Patients’ demographic data.

Parameter	PRP Group (10 Patient)	Control Group (10 Patients)	*p*-Value
Gender (M/F)	8/2	8/2	
Age * (O/M/F) (years)	58 (range; 46–63)/57/62	60.3 (range; 54–62)/60.6/59	0.186 **
Duration of symptoms * (O/M/F) (months)	6.8 (range; 4–12)/6.5/8	5.2 (range; 3–12)/5/6	0.217 **
VAS * (O/M/F)	6.6 (range; 3–8)/6.9/5.5	6.4 (range: 4–9)/6.5/6	0.355 **
Smoking (M/F)	2/0	2/0	
Hypertension (M/F)	2/0	2/0	
Dyslipidemia (M/F)	1/1	1/0	
Hyperthyroidism (M/F)	0	1/0	

* Mean value. ** Overall value. Abbreviations: F: females, M: males, O: overall, VAS: visual analogue scale for shoulder pain at the time of recruitment.

**Table 2 ijms-25-07957-t002:** The mean Bonar score in the experimental and control groups.

Scores	Experimental Group	Control Group	*p*-Value
Tenocyte morphology	0.1 (range; 0–1)	1.2 (range; 1–2)	0.001
Ground substance	0.1 (range; 0–1)	0.1 (range; 0–1)	1
Collagen architecture	0.1 (range; 0–1)	1.6 (range; 1–2)	0.002
Vascularity	0.1 (range; 0–1)	0.1 (range; 0–1)	1
Total	0.4 (range; 0–1)	3 (range; 2–4)	0.004

**Table 3 ijms-25-07957-t003:** The Bonar score [45].

Variables	Grade 0	Grade 1	Grade 2	Grade 3
Tenocyte morphology	Inconspicuous elongated spindle shaped nuclei with no obvious cytoplasm at light microscopy	Increased roundness: nucleus becomes more ovoid to round in shape without conspicuous cytoplasm	Increased roundness and size: the nucleus is round, slightly enlarged and a small amount of cytoplasm is visible	Nucleus is round, large with abundant cytoplasm and lacuna formation (chondroid change)
Ground substance	No stainable ground substance	Stainable mucin between fibers but bundles still discrete	Stainable mucin between fibers with loss of clear demarcation of bundles	Abundant mucin throughout with inconspicuous collagen staining
Collagen architecture	Collagen arranged in tightly cohesive well-demarcated bundles with a smooth dense bright homogeneous polarization pattern with normal crimping	Diminished fiber polarization: separation of individual fibers with maintenance of demarcated bundles	Bundle changes: separation of fibers with loss of demarcation of bundles giving rise to expansion of the tissue overall and clear loss of normal polarization pattern	Marked separation of fibers with complete loss of architecture
Vascularity	Inconspicuous blood vessels coursing between bundles	Occasional cluster of capillaries, less than one per 10 high-power fields	1–2 clusters of capillaries per 10 high power fields	Greater than two clusters per 10 high-power fields

## Data Availability

The data presented in this study are available on request from the corresponding author due to privacy reasons.

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
