# Peer review of "The Biological Effect of Platelet-Rich Plasma on Rotator Cuff Tears: A Prospective Randomized In Vivo Study"

_ijms, 2024, doi:10.3390/ijms25147957_

Round 1

Reviewer 1 Report (Previous Reviewer 2)

Comments and Suggestions for Authors

The revised manuscript can be accepted for publication. 

Comments on the Quality of English Language

Minor editing of English language required

Author Response

Dear Reviewers,

We would like to thank you for accepting to reconsider our manuscript titled: “The biological effect of platelet rich plasma on rotator cuff tear: A prospective randomized in vivo study” for publication in the International Journal of Molecular Sciences journal.

We would also like to thank you for the insightful comments. All raised points have been addressed and the manuscript has been revised according to their suggestions. All text changes in the manuscript have been highlighted. For reviewing purposes, the comments have been addressed one by one.

 Specifically:

Reviewer 1

Comment: Minor editing of English language required.

Reply: Thank you for your comment. The manuscript has been edited and re-checked for spelling and grammar errors and corrections have been made accordingly.

Reviewer 2 Report (Previous Reviewer 1)

Comments and Suggestions for Authors

Dear Editor and Authors,

Thank you for the opportunity to review the manuscript entitled “The biological effect of platelet rich plasma on rotator cuff tear: A prospective randomized in vivo study”. The Authors aimed to evaluate the effect of PRP on torn human supraspinatus tendon, by injecting it in the subacromial space six weeks prior to the scheduled arthroscopic surgery. Biopsy samples were harvested from patients injected and those not (control group). The specimens were evaluated under optical and electron microscopy. They found that the experimental group yielded abundant oval-shaped cells with multiple cytoplasmic processes within mainly parallel collagen fibers and less marked inflammation, simulating the intact tendon structure. The Authors concluded that these findings indicate that PRP can induce microscopic changes on the ruptured tendon by stimulating the healing process and facilitate a more effective recovery. 

 I have just some minor comments:

-       Add number of the examined participants from both groups in the Abstract

-       Intro “While the optimal platelet and leukocyte concentration is still debated,” – although this is true, I suggest you cited here the most commony used criteria for plasma to be “PRP”

-       Material and Method – what concentration of platelets can be obtained with your kit? What was a mean concentration of plt in PRP in your group?

Author Response

Dear Reviewers,

We would like to thank you for accepting to reconsider our manuscript titled: “The biological effect of platelet rich plasma on rotator cuff tear: A prospective randomized in vivo study” for publication in the International Journal of Molecular Sciences journal.

We would also like to thank you for the insightful comments. All raised points have been addressed and the manuscript has been revised according to their suggestions. All text changes in the manuscript have been highlighted. For reviewing purposes, the comments have been addressed one by one.

 Specifically:

Reviewer 2

Comment: Add number of the examined participants from both groups in the Abstract.

Reply: Thank you for your comment. Details about the number of patients in each group and randomization process have been added in the Abstract.

Comment: Intro “While the optimal platelet and leukocyte concentration is still debated,” – although this is true, I suggest you cited here the most commonly used criteria for plasma to be “PRP”.

Reply: Thank you for your comment. The criteria for PRP treatment according to the number of platelets have been inserted in the text.

Comment: Material and Method – what concentration of platelets can be obtained with your kit? What was a mean concentration of plt in PRP in your group?

Reply: Thank you for your comment. Based on manufacturer information, the TriCell PRP M Blood Separation Kit (REV-MED) can deliver 3 to 5 billion of platelets per injection. This information has been added in the text.

However, we did not send a sample of the produced PRP fluid for analysis and therefore we could not prove the exact concentration of platelets. We thought that any reduction of the small volume of the manufactured PRP material (3-4 ml) would affect its impact on tendon tissue. This information has been added in “limitations” part of Discussion.

This manuscript is a resubmission of an earlier submission. The following is a list of the peer review reports and author responses from that submission.

Round 1

Reviewer 1 Report

Comments and Suggestions for Authors

Dear Editor and Authors,

Thank you for the opportunity to review the manuscript entitled “The biological effect of platelet rich plasma on rotator cuff tear: A prospective randomized in vivo study”. The Authors aimed to evaluate the effect of PRP on torn human supraspinatus tendon, by injecting it in the subacromial space six weeks prior to the scheduled arthroscopic surgery. Biopsy samples were harvested from patients injected and those not (control group). The specimens were evaluated under optical and electron microscopy. They found that the experimental group yielded abundant oval-shaped cells with multiple cytoplasmic processes within mainly parallel collagen fibers and less marked inflammation, simulating the intact tendon structure. The Authors concluded that These findings indicate that PRP can induce microscopic changes on the ruptured tendon by stimulating the healing process and facilitate a more effective recovery. The ms however needs some revisions and specifications. 

-       In the Intro you should provide more information about PRP – definition, concentrations, and proven action related to concentrations. Also, other scientifically verified application of PRP should be mentioned

-       Results – “The two groups did not differ in respect to age, gender distribution, duration of symptoms and Visual Analogue Scale (VAS) score.” – here you should specify VAS scale /VAS scale regarding…what?/ - the same for table 1

-       In the Material and Methods – you should explain why you chose to do PRP 6 weeks before intervention? Provide a rationale and most important /!/: provide details about your PRP – how you obtained it? How you checked if it is a true PRP /validation of your protocol? a kit?/? What was a mean concentration of platelets? Did a concentration correlate with the effect?

-       Minor editing and proofing – English proofreading is needed, other e.g. dots instead of comas in tables for p

Author Response

  Dear Editor,

  We would like to thank you for accepting to reconsider our manuscript titled: “The biological effect of platelet rich plasma on rotator cuff tear: A prospective randomized in vivo study” for publication in the International Journal of Molecular Sciences journal.

  We would also like to thank the reviewers for their insightful comments. All raised points have been addressed and the manuscript has been revised according to their suggestions. All text changes in the manuscript have been highlighted. For reviewing purposes, the comments have been addressed one by one.

Reviewer 1

Comment: “In the Intro you should provide more information about PRP – definition, concentrations, and proven action related to concentrations. Also, other scientifically verified application of PRP should be mentioned”

Reply: Thank you for your comment. We have added a paragraph with information regarding the clinical applications of PRP and if different concentrations of PRP could affect the outcome of injection (Lines 72-80).

Comment: “Results – “The two groups did not differ in respect to age, gender distribution, duration of symptoms and Visual Analogue Scale (VAS) score.” – here you should specify VAS scale /VAS scale regarding…what?/ - the same for table 1”

Reply: Thank you for your comment. We have clarified that VAS was referred to “shoulder pain” (Line 112).

Comment: “In the Material and Methods – you should explain why you chose to do PRP 6 weeks before intervention? Provide a rationale and most important /!/: provide details about your PRP – how you obtained it? How you checked if it is a true PRP /validation of your protocol? a kit?/? What was a mean concentration of platelets? Did a concentration correlate with the effect?”

Reply: Thank you for your comment. We have provided details about the PRP preparation method (lines 330-337). We have inserted also a section describing the injection procedure and the reason for performing the PRP injection at 6 weeks before operative intervention (Lines 339-345). In addition, we have added a paragraph in the “Discussion”, where we analyze the effect of leukocyte rich- or -poor-PRP on tendon pathology and we justify the decision for choosing the leukocyte poor-PRP (Lines 255-266). In the current study, the PRP was prepared using the TriCell PRP M Blood Separation Kit (REV-MED). Based on the manufacturer’s instructions, the final product was leukocyte-poor PRP. However, we did not send a sample of the produced PRP fluid for analysis and therefore we could not prove the exact concentration of platelets. We thought that any reduction of the small volume of the manufactured PRP material (3-4 ml) might affect its efficacy in tendon pathology. 

Comment: “Minor editing and proofing – English proofreading is needed, other e.g. dots instead of comas in tables for p”

Reply: Thank you for your comment. The manuscript has been re-checked for spelling and grammar and corrections have been made accordingly. The issues with dots, comas and p values have been also addressed.

Reviewer 2 Report

Comments and Suggestions for Authors

Reviewing the present manuscript was a great pleasure for me. The topic is interesting for the readers of the journal, the text is well-written, and the quality of images is satisfactory. Likewise, some revisions are necessary before an eventual publication in the journal.

Keywords

Please, check that all the keywords are included in the MeSH database.

The progressive numeration of the different sections of the manuscript is wrong.

2. Results

3. Discussion

4. Materials and Methods

Please revise them as follows:

2.  Materials and Methods

3. Results

4. Discussion

Results

"In the control group areas of accumulated inflammatory cells indicated a more extended inflammation process."

In the rotator cuff tendons tear, the disrupted collagen fibers float inside the subacromial space. At this level, the synovial tissue, originating from the overlying bursa and the underlying glenohumeral joint, shows inflammatory processes secondary to abnormal frictions - i.e., focal synovitis. The peri-tendinous synovitis counteracts the normal healing process of the tendon tissue. In this sense, the inflammatory cells reported by the authors in the results of their study, "reach" the tendon tissue from the hyperplastic vessels of the subintimal layer of the synovial tissue secondary to hypertrophic synovitis. 

This histopathological aspect is pivotal to explaining the anti-inflammatory effect of the PRP and subsequently its regenerative effect. Indeed, the ability of PRP to reduce focal synovitis encasing the injured tendon is a pivotal step in promoting a secondary healing process of the tendon tissue. 

For a comprehensive description of this histopathological mechanism, please refer to Pathol Res Pract. 2023 Jan;241:154273. doi: 10.1016/j.prp.2022.154273. Epub 2022 Dec 12. PMID: 36563558.

Minor corrections

- I suggest the authors clearly specify throughout the manuscript that the injection of PRP was not blind but performed under ultrasound guidance

Once again, the reduced inflammatory process of the free end of the torn tendon (secondary to a reduced peri-tendinous synovitis) is a crucial point to optimize the post-surgical outcome of tendon repair. This topic should be clearly described in this section of the manuscript.

Limitations

What about the follow-up of patients using high-resolution ultrasound imaging? Among the limitations of this investigation, the authors should add the lack of ultrasound assessment of tendon tissue after the PRP injection (e.g., the texture of the tendon, hyperemia, and hypervascularization with color/power Doppler, dynamic scans to assess the spatial extension of the injury, etc.).

Materials and Methods

- the serial number of the Ethics Committee must be added to this section of the manuscript

Study design

"Patients of the experimental group underwent an ultrasound-guided autologous PRP injection in the subacromial space".

Considering the use of ultrasound guidance, I suppose the PRP has been injected near the tendon injury. I suggest the authors add a sonographic image of the procedure to optimize the reproducibility of this investigation by the readers. 

Please, better specify the technical features of the ultrasound-guided procedure. In-plane technique? Lateral-to-medial approach? Etc.

4.2 Patient selection

"Patients with supraspinatus tendon tear, who would be treated with arthroscopic repair, were deemed eligible to participate in the current study".

How the diagnosis of supraspinatus tendon tear was provided? Ultrasound imaging? MRI? Please, better specify this aspect in the text.

Why a low count of platelets on blood examination was not considered among the exclusion criteria?

4.3 Tendon sample harvesting

Can the authors add an arthroscopic image that shows the site of sampling exactly? The free end of the torn tendon that floats inside the subacromial space?

Comments on the Quality of English Language

Minor editing of English language required. 

Author Response

  Dear Editor,

  We would like to thank you for accepting to reconsider our manuscript titled: “The biological effect of platelet rich plasma on rotator cuff tear: A prospective randomized in vivo study” for publication in the International Journal of Molecular Sciences journal.

  We would also like to thank the reviewers for their insightful comments. All raised points have been addressed and the manuscript has been revised according to their suggestions. All text changes in the manuscript have been highlighted. For reviewing purposes, the comments have been addressed one by one.

Reviewer 2

Comment: “Keywords. Please, check that all the keywords are included in the MeSH database.”

Reply: Thank you for your comment. We have revised the keywords in order to be included in MeSH database (Line 32).

Comment: “The progressive numeration of the different sections of the manuscript is wrong. 2. Results 3. Discussion 4. Materials and Methods. Please revise them as follows: 2.  Materials and Methods 3. Results 4. Discussion”

Reply: Thank you for your comment. We totally agree with the above instruction but the order of the sections of the article has been prepared in consistent with the manuscript template of the Journal (IJMS). If the Editor agrees we can change the order of the sections accordingly.

Comment: “Results. "In the control group areas of accumulated inflammatory cells indicated a more extended inflammation process." In the rotator cuff tendons tear, the disrupted collagen fibers float inside the subacromial space. At this level, the synovial tissue, originating from the overlying bursa and the underlying glenohumeral joint, shows inflammatory processes secondary to abnormal frictions - i.e., focal synovitis. The peri-tendinous synovitis counteracts the normal healing process of the tendon tissue. In this sense, the inflammatory cells reported by the authors in the results of their study, "reach" the tendon tissue from the hyperplastic vessels of the subintimal layer of the synovial tissue secondary to hypertrophic synovitis. This histopathological aspect is pivotal to explaining the anti-inflammatory effect of the PRP and subsequently its regenerative effect. Indeed, the ability of PRP to reduce focal synovitis encasing the injured tendon is a pivotal step in promoting a secondary healing process of the tendon tissue. For a comprehensive description of this histopathological mechanism, please refer to Pathol Res Pract. 2023 Jan;241:154273. doi: 10.1016/j.prp.2022.154273. Epub 2022 Dec 12. PMID: 36563558.”

Reply: Thank you very much for your useful comment. All this information has been added  in the discussion section (Lines 180-181) and the suggested reference has been also inserted in the relevant list.

Comment: “I suggest the authors clearly specify throughout the manuscript that the injection of PRP was not blind but performed under ultrasound guidance”

Reply: Thank you for your comment. More information about the procedure of PRP injection has been added in the manuscript (Lines 327-333)

Comment: “Once again, the reduced inflammatory process of the free end of the torn tendon (secondary to a reduced peri-tendinous synovitis) is a crucial point to optimize the post-surgical outcome of tendon repair. This topic should be clearly described in this section of the manuscript.”

Reply: Thank you again for your comment and information. According to your instructions, we have added this topic and we have modified  the Discussion section accordingly (Lines 194-196)

Comment: “What about the follow-up of patients using high-resolution ultrasound imaging? Among the limitations of this investigation, the authors should add the lack of ultrasound assessment of tendon tissue after the PRP injection (e.g., the texture of the tendon, hyperemia, and hypervascularization with color/power Doppler, dynamic scans to assess the spatial extension of the injury, etc.).”

Reply: Thank you for your comment. The lack of post-intervention ultrasound assessment is considered one drawback of the study and this information has been added in “limitations” paragraph of Discussion section (Line 296-299).

Comment: “the serial number of the Ethics Committee must be added to this section of the manuscript”

Reply: Thank you for the comment.  The serial number of the Ethics committee Approval has been inserted in the text (Line 303).

Comment: “Patients of the experimental group underwent an ultrasound-guided autologous PRP injection in the subacromial space". Considering the use of ultrasound guidance, I suppose the PRP has been injected near the tendon injury. I suggest the authors add a sonographic image of the procedure to optimize the reproducibility of this investigation by the readers.”

Reply: Thank you for the comment.  The PRP fluid was injected into the subacromial space of the examined shoulder under ultrasound guidance. This information along with a relevant image have been added in the manuscript (Line 347).

Comment: “Please, better specify the technical features of the ultrasound-guided procedure. In-plane technique? Lateral-to-medial approach? Etc.”

Reply: Thank you for your comment. The ultrasound-guided procedure  of PRP application is presented in a better way at the text and a relevant picture has been added also (Lines 339-347).

Comment: “"Patients with supraspinatus tendon tear, who would be treated with arthroscopic repair, were deemed eligible to participate in the current study". How the diagnosis of supraspinatus tendon tear was provided? Ultrasound imaging? MRI? Please, better specify this aspect in the text.”

Reply: Thank you for your comment. The diagnosis of supraspinatus tear was confirmed by MRI. According to your recommendation this issue is presented more specifically in the manuscript (Lines 320-321)

Comment: “Why a low count of platelets on blood examination was not considered among the exclusion criteria?”

Reply: Thank you for your comment. As all participants were relatively healthy regarding not only the platelets count or any blood disorders but also other severe medical disorders, we ignored to report this issue as one of the exclusion criteria. However, we totally agree with the comment and the existence of  low platelets number has been inserted in the text as one of the main exclusion criteria of the study (Lines 327-328).

Comment: “Can the authors add an arthroscopic image that shows the site of sampling exactly? The free end of the torn tendon that floats inside the subacromial space?”

Reply: Thank you for your comment. An arthroscopic image of tendon harvesting procedure has been added in the manuscript (Line 361).

Round 2

Reviewer 1 Report

Comments and Suggestions for Authors

All comments were addressed and all questions answered. Thank you.

Author Response

Thank you for the review of our article

Reviewer 2 Report

Comments and Suggestions for Authors

The revised manuscript can be accepted for publication.

Please, pay attention to the progressive numeration of the different sections of the manuscript:

1. Introduction

2. Materials and Methods

3. Results

4. Discussion

5. Conclusion

Comments on the Quality of English Language

Minor editing of English language required. 

Author Response

  Dear Editor and Reviewers,

  We would like to thank you for accepting to reconsider our manuscript titled: “The biological effect of platelet rich plasma on rotator cuff tear: A prospective randomized in vivo study” for publication in the International Journal of Molecular Sciences journal.

  We would also like to thank the reviewers for their insightful comments. All raised points have been addressed and the manuscript has been revised according to their suggestions. All text changes in the manuscript have been highlighted. For reviewing purposes, the comments have been addressed one by one.

Reviewer 2

Comment: “Minor editing of English language required”

Reply: Thank you for your comment. A comprehensive and detailed manuscript editing was undertaken.

Comment: “Please, pay attention to the progressive numeration of the different sections of the manuscript..”

Reply: Thank you for your comment. According to your instructions, we have changed the order of the sections of the article (1. Introduction 2. Materials and Methods 3. Results 4. Discussion 5. Conclusion)